# I Want to Be You(r Friend): An Investigation of the Effects of Gendered Personality Traits on Engagement with Different *Modern Family* Characters

**Bartosz G. Żerebecki ***, Esther van der Vliet and Julia Kneer

Department of Media and Communication, Erasmus School of History, Culture and Communication, Erasmus University Rotterdam, 3062 PA Rotterdam, The Netherlands; esther_vdvliet@live.nl (E.v.d.V.); kneer@eshcc.eur.nl (J.K.)
* Correspondence: zerebecki@eshcc.eur.nl

**Abstract:** The extant research focused on gender understood as a single item to explain wishful identification and parasocial relationships with TV characters. This study focused on gendered personality traits and how they contribute to wishful identification, parasocial relationships with (non-)stereotypical male and female characters of the TV series *Modern Family*, and the series enjoyment in general. Participants (N = 508) were randomly assigned to one of four conditions with questions about either stereotypical male or female or non-stereotypical male or female characters. Respondents also answered questions about their own gender traits (positive/negative feminine/masculine), wishful identification, parasocial friendship and love for the assigned character, and enjoyment of the series. Different gendered personality traits were associated with wishful identification, parasocial friendship, and love with different types of characters, as well as series enjoyment. Thus, we conclude that media characters should exhibit both stereotypical and non-stereotypical traits to reach a broad and diverse audience.

**Keywords:** gender traits; parasocial interactions; intersectionality

## 1. Introduction

TV fans often want to act like or be friends with their media idols, even when these idols are fictional characters (Hoffner and Buchanan 2005; Rosaen and Dibble 2017; Tukachinsky 2010). Moreover, some viewers fall in love with TV show characters (Tukachinsky 2010). Still, the fact that people create meaningful relationships with media personas is nothing new. Already in the 1950s, Horton and Wohl (1956) had coined the term "parasocial interactions" to describe the one-sided relationships between viewers and onscreen characters. These relationships have continued to attract scholarly attention ever since (e.g., Levy 1979; Perse and Rubin 1989; Tukachinsky 2010). Thanks to such scholarship, we know that stronger relationships develop when viewers are somehow similar to characters, for example, in terms of age, gender, or attitudes (Bui 2017; Hoffner and Buchanan 2005; Steinke et al. 2012). However, in an increasingly diverse media landscape, with characters representing a multitude of identities (e.g., racial, ethnic, gender, and sexual, among others), it could be difficult to find such simple demographic dimensions of similarity. Thus, to understand better how relationships with fictional characters take place, some researchers call for more exploration of different ways in which viewers can see themselves as similar to media personas (Żerebecki et al. 2021).

These previously identified dimensions of similarity need to be reconceptualized in order to capture more diversity among people. As a first step in this process, we zoom in on gender identity because of its complexity, which includes different behaviors and personality traits (Wood and Eagly 2009). Additionally, our choice was motivated by the existence of a variety of measuring instruments that capture different aspects of gender

that are sometimes overlooked in research (Bem 1974; Berger and Krahé 2013; Reich 2021). Simple categories, such as male, female, non-binary, or transsexual, often fail to capture full gender diversity (Whyte et al. 2018). Still, such conceptualizations are widely used in quantitative research (Bauer et al. 2017; Kneer et al. 2019; Reich 2021). We argue that a more robust conceptualization of gender identity enables a better understanding of possible similarities between audiences and onscreen characters. Possibly, viewers can see aspects of themselves in various characters, despite differently labeled genders, due to a similarity of behaviors, emotions, and attitudes associated with gender identities. We acknowledge that other identity aspects, such as age, ethnicity, race, sexual orientation, and class, could also be reconceptualized to include possible new ways of being familiar despite perceived differences. We hope to inspire more research in this area by exploring gender and its different measurements in depth.

To achieve a more nuanced understanding of gender, we apply the intersectionality paradigm, which traditionally examines the unique experiences of people with various overlapping identities, such as gender, race, and sexuality (Mays and Ghavami 2018). However, intersectionality also "invites us to approach the study of social categories with more complexity and suggests ways to bring more nuance and context to our research on the social categories" (Cole 2009, p. 179). Thus, in this research, we zoom in on gender identity in its multiple iterations to follow Cole's (2009) call to expand research with intersectional lenses. Specifically, we focus on gendered personality traits, which include various dispositions grouped as positive femininity (e.g., being social), negative femininity (e.g., being anxious), positive masculinity (e.g., being rational), and negative masculinity (e.g., being arrogant) (Berger and Krahé 2013). Thus, we capture a nuanced understanding of gender identity that moves from a single-item social category to a description of an individual on different dimensions (Reich 2021). Such an approach to gender allows us to see audience members in an intersectional manner, i.e., as complex humans where different aspects of gender identity overlap to produce unique experiences. This is especially relevant in our exploration of similarity with media characters. One viewer could have traits classified as masculine and feminine and positive and negative at the same time and thus relate more to either male or female TV show characters, depending on their personality and gender identity. Importantly, gender identity overlaps with other social categories, such as race, ethnicity, sexual orientation, religion, and class, which brings out even more nuances in people. However, a full account of such complexity falls outside the scope of our article. Thus, we focus on one identity aspect to ensure a sufficient depth of analysis.

Some research has already explored how individuals with different gendered personalities use media such as video games (Kneer et al. 2019; Reich 2021). However, we still do not know how this conceptualization of gender relates to engagement with TV show characters. TV shows, and especially sitcoms, have a long tradition of portraying male and female characters that behave in a socially expected, often stereotypical way (Walsh et al. 2008; Wille et al. 2018). Notably, a slow shift to more diverse portrayals in entertainment TV is happening (Sink and Mastro 2017). Characters are increasingly portrayed with personality traits, professions, and interests that transcend expectations and older stereotypes. An example of this move could be *Modern Family* (Lloyd and Levitan 2009–2020), an ABC TV series that features male and female characters that affirm and break social expectations (Buchbinder 2014; Mora 2018). In our research, we explore engagement with different *Modern Family* characters in relation to different gendered personality traits. Such focus allows us to dive deeper into the exploration of possible ways in which audience members could find themselves similar to characters based on personalities.

In short, the goal of this research is twofold: (1) to verify that gendered personality traits are valuable predictors when studying relationships with TV characters and (2) to analyze how the characters' (non-)stereotypical gender portrayal influences the relationships between audience members and media figures. To fulfill these goals, we conducted a survey-experiment where each participant self-reported classification according to four gendered personality dimensions (i.e., positive femininity, negative femininity, positive masculinity, and negative masculinity). Then, the respondents were randomly exposed to one of four different characters from *Modern Family* that either proved or disproved expectations about men and women's personality traits. Finally, we measured engagement with the presented media persona (i.e., wishful identification, parasocial friendship, and parasocial love) and general TV show enjoyment to see relations between gendered personality traits and various engagement measurements.

## 2. Literature Review

In this section, we define and discuss different gendered personality traits and media portrayals of men and women. Then, we zoom in on engagement with media characters and discuss pertinent theories and research that show the importance of perceived similarity to pose hypotheses about relationships between gendered personality traits and wishful identification; parasocial interactions, including friendships and love; and enjoyment of TV shows.

### 2.1. Viewers' Gender Traits and Characters' Gender Stereotypes

As mentioned before, gender identity is a multifaceted construct (Berger and Krahé 2013; Whyte et al. 2018; Wood and Eagly 2009). Berger and Krahé (2013) updated an earlier take by Bem (1974) on in-depth measurements of masculinity and femininity. They argued that identity includes both desirable and undesirable characteristics. In a series of quantitative studies, which involved surveys with young Germans, they demonstrated four different dimensions of gender: positive femininity, negative femininity, positive masculinity, and negative masculinity. Positive femininity means being emotional, loving, sensitive, and emphatic, while negative femininity captures being anxious, oversensitive, and self-doubting. In contrast, positive masculinity involves being logical, practical, and rational, while negative masculinity embraces being arrogant, power-hungry, boastful, and inconsiderate. This understanding of gender identity allows each surveyed individual to report unique scores on each personality aspect, thus providing a more complex view of identity when compared to a simple identification as being male, female, or non-binary, among others.

In contrast to the diversity of real people and their personalities, media characters often embody more fixed identities that become stereotypical (Walsh et al. 2008; Wille et al. 2018). Gender stereotypes are set beliefs about expected, proper behaviors, roles, and traits for men and women (Wille et al. 2018). For instance, men are perceived to be more driven by dominance, agency, independence, and competence, while women are perceived to be more driven by community, expressivity, and emotionality (Haines et al. 2016). These gender stereotypes are similar to different aspects of positive and negative masculinity and femininity. Both positive and femininity negative femininity include being sensitive, emotional, and more community oriented, with differences in degrees of these orientations. Similarly, both positive masculinity and negative masculinity involve being independent, self-reliant, and self-confident, with similar differences in the intensity of these dispositions. Media characters often have set personality traits and ways of behaving that typify one particular identity as displayed by *Modern Family* characters (Buchbinder 2014). In contrast, the audience members are real people that display a rich variety of different personalities and behaviors. Thus, it is likely that different viewers can relate to different characters that both confirm and disprove stereotypes depending on their own gender identity.

*2.2. Media Engagement Theories*

Engagement with TV shows and their characters includes multiple different psychological processes. First, we describe general theories that explain when engagement with media is greater. Then, we discuss various types of media engagement studied in this research, namely wishful identification, parasocial friendship and love, and enjoyment of the TV show.

In their seminal research, Perse and Rubin (1989), based on a survey of American college students, argued that relationships with soap opera characters work in a similar way to relationships with real people. They posed that stronger bonds are built when a viewer can predict characters' actions, which stems from the psychological need for uncertainty reduction. Therefore, familiar characters are likely to elicit greater viewers' engagement. This line of thinking has developed over time. Understanding media characters can also stem from similarities to them. In a recent literature review, Żerebecki et al. (2021) pointed out multiple studies that demonstrate the importance of similarity between viewers and media characters for media engagement. Oftentimes, such similarity is conceived of in terms of demographics. For instance, Bui (2017) found that people tend to choose a favorite media celebrity of the same age and gender. Still, it is possible that viewers can find different ways in which they find themselves similar to media characters, such as personality, behaviors, and attitudes. In short, the extant research indicates that media engagement is related to a sense of familiarity and similarity with onscreen characters.

2.2.1. Wishful Identification

Wishful identification is defined as wanting to emulate a media persona in terms of behaviors and attitudes (Hoffner and Buchanan 2005). Keeping in mind that uncertainty reduction is important for developing bonds with media characters (Perse and Rubin 1989), we argue that gender stereotypes could help to increase a sense of familiarity and could make it easier to understand the media characters. Additionally, since men still enjoy a privileged media representation, they are often portrayed as central characters in TV shows (Wille et al. 2018; Wood and Eagly 2009). Such representation could influence all audience members despite their gender to want to be similar to the main hero of the story. Thus, we pose our first hypothesis:

**H1.** *Wishful identification is higher for (a) stereotypical characters and (b) male characters.*

The previous research found that levels of wishful identification with a given character depend on levels of similarity with the character (Hoffner and Buchanan 2005; Steinke et al. 2012). Hoffner and Buchanan (2005) pointed out the importance of attitudinal similarity, which encompasses a general sense of being similar to the character. Steinke et al. (2012) reported the importance of shared gender. Based on these findings, we argue that similarity in terms of personality also predicts wishful identification. Stereotypical female characters are more community oriented, expressive, and emotional, which aligns with both positive (i.e., being emotional and empathic) and negative (i.e., being oversensitive) femininity. Additionally, stereotypical male characters are thought to be driven by agency and independence, which is similar to both positive (i.e., being practical) and negative masculinity (i.e., being power-hungry). Analogically, we posit that non-stereotypical characters are most similar to people exhibiting higher scores on opposite gendered personality traits (i.e., non-stereotypical female characters are similar to people with positive and negative masculinities, and vice versa). Finally, we assume that lack of similarity implies lower wishful identification, because of the stereotypical traits being opposite to the gendered personality traits. Thus, we pose the following hypotheses:

**H2.** *Wishful identification with stereotypical female characters increases with (a) positive femininity and (b) negative femininity and decreases with (c) positive masculinity and (d) negative masculinity.*

**H3.** *Wishful identification with non-stereotypical female characters decreases with (a) positive femininity and (b) negative femininity and increases with (c) positive masculinity and (d) negative masculinity.*

**H4.** *Wishful identification with stereotypical male characters decreases with (a) positive femininity and (b) negative femininity and increases with (c) positive masculinity and (d) negative masculinity.*

**H5.** *Wishful identification with non-stereotypical male characters increases with (a) positive femininity and (b) negative femininity and decreases with (c) positive masculinity and (d) negative masculinity.*

### 2.2.2. Parasocial Relationships

Horton and Wohl (1956) described parasocial relationships as "a one-sided interpersonal relationship that television viewers establish with media characters" (p. 280). Parasocial relationships can become friendships, where audience members feel that they understand a mediated character equally well as a real-life friend (Perse and Rubin 1989). Furthermore, Tukachinsky (2010) proposed that there are two types of parasocial relationships: friendly and romantic ones. A romantic parasocial relationship is built on the same experiences as parasocial friendships but is stronger in intensity and more akin to love. Originally, Tukachinsky (2010) argued for the inclusion of both physical and emotional attraction. However, we decided to focus on emotional attraction because of its closeness to parasocial friendships. To pose the next hypothesis on parasocial friendship, we follow a similar logic on characters' familiarity and stereotypical representation as in the wishful identification section. Moreover, in a metanalysis, Tukachinsky et al. (2020) reported nine studies that showed parasocial relationships to be stronger with characters perceived as attractive. As mentioned, men are often portrayed more attractively on TV (Wille et al. 2018; Wood and Eagly 2009). Thus, we propose that:

**H6.** *Parasocial friendship is higher for (a) stereotypical characters and (b) male characters.*

Analogically to wishful identification, parasocial relationships are stronger with similar characters (Cohen and Hershman-Shitrit 2017; Tukachinsky et al. 2020). Tukachinsky et al. (2020) reported 16 studies showing this relationship. Moreover, Cohen and Hershman-Shitrit (2017) empirically demonstrated that personality similarity can predict feeling closer to a media character. Another important predictor of stronger parasocial relationships could be wishful identification (Hu et al. 2021; Lim et al. 2020). Logically, wanting to be similar to someone implies a level of attraction, which could lead to wanting to be friends. Therefore, following the previous discussion of similarity between stereotypical characters and gendered personality traits and the reported research, we pose these hypotheses:

**H7.** *Parasocial friendship with stereotypical female characters increases with (a) positive femininity, (b) negative femininity, and (c) wishful identification and decreases with (d) positive masculinity and (e) negative masculinity.*

**H8.** *Parasocial friendship with non-stereotypical female characters decreases with (a) positive femininity and (b) negative femininity and increases with (c) positive masculinity, (d) negative masculinity, and (e) wishful identification.*

**H9.** *Parasocial friendship with stereotypical male characters decreases with (a) positive femininity and (b) negative femininity and increases with (c) positive masculinity, (d) negative masculinity, and (e) wishful identification.*

**H10.** *Parasocial friendship with non-stereotypical male characters increases with (a) positive femininity, (b) negative femininity, and (c) wishful identification and decreases with (d) positive masculinity and (e) negative masculinity.*

As argued before, emotional parasocial love and parasocial friendship differ in intensity, with the former being stronger than the latter. Therefore, for parasocial love, we pose analogical hypotheses to the ones about parasocial friendships:

**H11.** *Parasocial love is higher for (a) stereotypical characters and (b) male characters.*

**H12.** *Parasocial love with stereotypical female characters increases with (a) positive femininity, (b) negative femininity, and (c) wishful identification and decreases with (d) positive masculinity and (e) negative masculinity.*

**H13.** *Parasocial love with non-stereotypical female characters decreases with (a) positive femininity and (b) negative femininity and increases with (c) positive masculinity, (d) negative masculinity, and (e) wishful identification.*

**H14.** *Parasocial love with stereotypical male characters decreases with (a) positive femininity and (b) negative femininity and increases with (c) positive masculinity, (d) negative masculinity, and (e) wishful identification.*

**H15.** *Parasocial love with non-stereotypical male characters increases with (a) positive femininity, (b) negative femininity, and (c) wishful identification and decreases with (d) positive masculinity and (e) negative masculinity.*

2.2.3. Enjoyment

Besides wanting to be similar to media characters or feeling friendly toward or in love with them, viewers can also enjoy the TV show in general. Previous research has posed that parasocial relationships with many TV show characters could be a source of media enjoyment (Baldwin and Raney 2021; Kim and Sintas 2021). Furthermore, Rosaen and Dibble (2017) explained that the most enjoyable parasocial relationships are those with characters similar to the viewer. Since *Modern Family* has a mix of stereotypical and counter-stereotypical characters, we believe that people with all the different gendered personality traits could find similar characters and, thus, enjoy the TV show. Therefore, we pose the following hypothesis:

**H16.** *TV show enjoyment increases with the experience of (a) positive femininity, (b) negative femininity, (c) positive masculinity, and (d) negative masculinity.*

**3. Materials and Methods**

*3.1. Procedure*

The data presented in this paper were used in a master thesis of the second author of this study. We collected the data using an online survey with an experimental design. This methodological choice allowed us to test the cited theories and the following hypotheses. Such survey design resulted in a three-part questionnaire. The first part asked the participants general questions about their viewing behavior of *Modern Family*. Only participants that knew the show were included in the survey. Before the second part of the survey, we showed participants a picture of one of the four selected characters, which represented either a stereotypical female (Haley), a non-stereotypical female (Alex), a stereotypical male (Luke), or a non-stereotypical male (Manny) character. The second part of the survey measured relationships with TV characters, focusing on the respondent's parasocial relationship and wishful identification with the presented character. Participants only answered questions about one assigned character. The third part of the survey asked respondents about their perceptions of their own gendered personality traits. This section was followed by questions about the demographics of the respondents. We tested the survey to remove any unclear phrasing before distributing it to a larger audience.

### 3.2. Sample

The sample consisted of N = 508 respondents. In the sample, there were 402 (79.1%) female respondents, 91 (17.9%) male respondents, and 12 (2.4%) non-binary or third-gender respondents, while 3 (0.6%) participants preferred to leave the question blank. The average age was 27.03 years (*SD* = 8.51), with the eldest respondent being 78 and the youngest being 18. Participants lived across 52 countries. Most of them reported living in the Netherlands (40.7%), the United States of America (15.4%), and the United Kingdom of Great Britain and Northern Ireland (10.7%). The majority of the participants had completed a bachelor's degree (42.3%), followed by respondents with some college experience but no degree (16.7%), a high school diploma (16.5%), and a master's degree (15%). Participants had seen on average 207.96 (*SD* = 67.53) episodes of *Modern Family*. The show had had 11 seasons and was 250 episodes long at the time of the study. The respondent who had seen the least number of episodes had watched six episodes. A total of 289 (56.9%) respondents had watched the entire show.

### 3.3. Stimulus Material

The TV show *Modern Family* was used as an exemplary and successful sitcom to test the proposed hypotheses. The show has enjoyed considerable commercial success among critics and audiences alike, with 85 Emmy nominations and 22 wins (Emmys n.d.; Rosen 2020). In the series, three related families and their struggles with everyday life in Los Angeles are shown (ABC n.d.). *Modern Family* has a diverse set of characters (Buchbinder 2014; Mora 2018), making it suitable for analyzing the effect of gendered personality traits on parasocial relationships and wishful identification with gender-stereotypical and gender-non-stereotypical televised characters. The second and third authors are fans of the TV show and based on their knowledge of the sitcom, they selected four characters as the stimulus material for the experimental design of this study. The decision was supported by Buchbinder's (2014) analysis of male characters and the designations of stereotypical male and female traits (Haines et al. 2016). These characters were Luke Dunphy, a stereotypical male character, Manny Delgado, a non-stereotypical male character, Haley Dunphy, a stereotypical female character, and Alex Dunphy, a non-stereotypical female character. A (non-)stereotypical gender portrayal means that the characters have various but sometimes similar personality traits. Luke is often shown as playful, practical, and independent, while Alex is portrayed to be smart, logical, and sometimes harsh. All these traits can be seen as analogical to both negative (through being self-centered and boastful) and positive masculinity (through being rational and solution oriented). In contrast, Manny is often shown to be caring, passionate, and sensitive, while Haley is depicted as a popular and friendly character. These traits are related more to both positive (through community orientation) and negative femininity (through being oversensitive and self-doubting). The four characters were selected to enable comparison between largely equivalent personas. They share relatively equal young ages and life situations but differ strongly in personality. Notably, our choice of characters was not perfectly equivalent as Manny is Latino and other characters are white (we elaborate on this limitation in the discussion section). However, we could not find other equivalent characters. For instance, older characters have different life situations, which would make the comparison between them harder.

### 3.4. Measures

*Gendered personality.* We assessed this concept via the scale by Berger and Krahé (2013; 5-point Likert scale; 1 = describes me extremely well and 5 = does not describe me). For the analysis, we reverse-coded the scale items so that a higher score indicated that the personality trait described the respondent more. Example items included "I would describe myself as analytical." All four subscales were found to be reliable (positive femininity Cronbach's $\alpha$ = 0.79; negative femininity Cronbach's $\alpha$ = 0.73; positive masculinity Cronbach's $\alpha$ = 0.79; negative masculinity Cronbach's $\alpha$ = 0.72).

*Wishful identification.* We measured the concept with three items using a 5-point Likert scale (1 = strongly agree, 5 = strongly disagree) based on Hoffner's (1996) scale. For the analysis, we reverse-coded the scale items so that a higher score indicated higher wishful identification. An example item included was "I wish I could be more like the character" (Cronbach's $\alpha$ = 0.80).

*Parasocial relationships.* We used the scale by Tukachinsky (2010), dividing the concept into parasocial friendship and parasocial love. Respondents answered 20 questions based on a 5-point Likert scale (1 = strongly agree, 5 = strongly disagree) about their assigned character. For the analysis, we reverse-coded the scale items so that a higher score indicated a higher level of parasocial relationships. Parasocial friendship consisted of 13 items related to parasocial friendship communication and parasocial friendship support (Cronbach's $\alpha$ = 0.88), for instance, "I think they could be my friend and if they were a real person. "For parasocial love, we only included seven items pertaining to emotional bonds (Cronbach's $\alpha$ = 0.89), for example, "I want them physically, emotionally, and mentally."

*Enjoyment.* Respondents were asked if they enjoy watching *Modern Family* in three 5-point Likert scale items from the Oliver and Bartsch (2010) scale (1 = strongly agree, 5 = strongly disagree; Cronbach's $\alpha$ = 0.93). For the analysis, we reverse-coded the scale items so that a higher score indicated higher enjoyment of the series. An example item was "It is fun for me to watch *Modern Family*."

*Demographics.* Respondents were asked about their gender affiliation (male, female, non-binary/third gender, or prefer not to say), their birth year, their current country of residence, and their highest achieved level of education.

*Familiarity.* Respondents were asked about their familiarity with the show by asking them for an estimation of how many episodes of *Modern Family* they had seen.

## 4. Results

### 4.1. Gendered Personality Traits and Wishful Identification

A 2 (male character vs. female character) $\times$ 2 (stereotypical character vs. non-stereotypical character) ANOVA was conducted with wishful identification as a dependent variable. ANOVA revealed a significant main effect for the gender of the character, $F(1, 504) = 10.89$, $p = 0.001$, $\eta^2_p = 0.02$, with participants experiencing more wishful identification with female characters than with male characters. A significant main effect for the stereotypicality of the character was also found, $F(1, 504) = 9.32$, $p = 0.002$, $\eta^2_p = 0.02$, with participants wishfully identifying more with non-stereotypical characters than with stereotypical characters. A significant interaction effect was also found combining the effect of the gender of the character and the stereotypicality of the character on wishful identification, $F(1, 504) = 9.77$, $p = 0.002$, $\eta^2_p = 0.02$.

Since we found a significant interaction effect, we investigated differences between specific subgroups. Looking first at the comparisons within (non-)stereotypical characters, post hoc *t*-tests revealed that participants experienced more wishful identification with the non-stereotypical female character than they did with the non-stereotypical male character, $t(251) = 4.53$, $p < 0.001$. No significant difference in wishful identification was found between stereotypical male and stereotypical female characters, $t(221, 1) = 0.12$, $p = 0.904$. Further comparing wishful identification specifically for just female characters, a post hoc *t*-test revealed that participants experienced more wishful identification with a non-stereotypical female character than with a stereotypical female character, $t(263) = 4.62$, $p < 0.001$. When comparing differences within male characters, no significant difference was found between the experience of wishful identification with stereotypical male and non-stereotypical male characters, $t(241) = 0.05$ $p = 0.961$. Based on the unexpected negative direction of the association found, H1a and H1b were both declined. Against our expectations, wishful identification is lower for stereotypical and male characters. The summary of these results is presented graphically in Figure 1 and in Table 1.

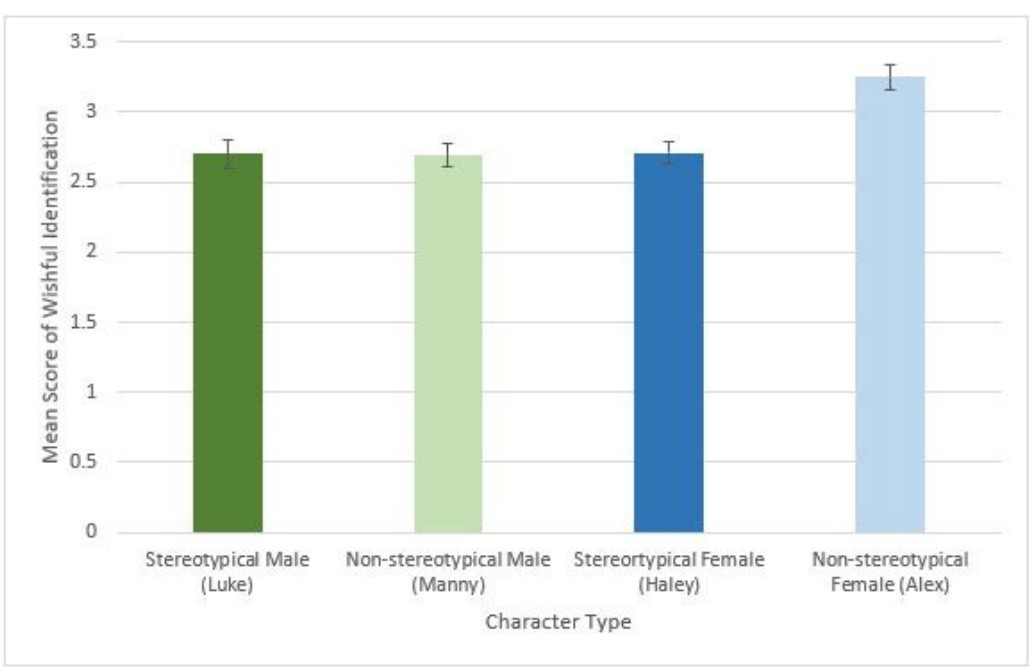

**Figure 1.** Mean scores of wishful identification with each type of character. *Note.* Error bars represent the standard error for mean scores.

**Table 1.** Means and standard deviations of wishful identification with various character groups.

|  | **Mean** | **Standard Deviation** |
|---|---|---|
| Stereotypical characters (Haley and Luke) | 2.71 | 0.97 |
| Non-stereotypical characters (Alex and Manny) | 2.97 | 1.02 |
| Male characters (Luke and Manny) | 2.70 | 1.01 |
| Female characters (Alex and Haley) | 2.96 | 0.98 |
| Stereotypical female character (Haley) | 2.71 | 0.89 |
| Non-stereotypical female character (Alex) | 3.25 | 0.99 |
| Stereotypical male character (Luke) | 2.70 | 1.06 |
| Non-stereotypical male character (Manny) | 2.69 | 0.96 |

For each of the four *Modern Family* characters, one multiple regression analysis was conducted using wishful identification as a criterion. Predictors were the gendered personality traits. A summary of this analysis can be found in Table 2.

**Table 2.** Standardized beta coefficients for multiple regression analyses predicting wishful identification with studied characters.

| Stereotypical female | |
|---|---|
| (≠H2a) Positive femininity | 0.16 |
| (=H2b) Negative femininity | 0.20 * |
| (≠H2c) Positive masculinity | −0.05 |
| (≠H2d) Negative masculinity | 0.06 |
| | $R^2 = 0.10$ |
| | $F(4, 134) = 3.91, p = 0.005$ |
| Non-stereotypical female | |
| (≠H3a) Positive femininity | 0.02 |
| (≠H3b) Negative femininity | 0.23 * |
| (≠H3c) Positive masculinity | 0.03 |
| (≠H3d) Negative masculinity | −0.12 |
| | $R^2 = 0.07$ |
| | $F(4, 119) = 2.14, p = 0.080$ |
| Stereotypical male | |
| (≠H4a) Positive femininity | −0.15 |
| (≠H4b) Negative femininity | 0.17 |
| (≠H4c) Positive masculinity | 0.13 |
| (≠H4d) Negative masculinity | 0.02 |
| | $R^2 = 0.05$ |
| | $F(4, 109) = 1.31, p = 0.271$ |
| Non-stereotypical male | |
| (≠H5a) Positive femininity | 0.06 |
| (≠H5b) Negative femininity | 0.11 |
| (≠H5c) Positive masculinity | −0.10 |
| (≠H5d) Negative masculinity | 0.04 |
| | $R^2 = 0.03$ |
| | $F(4, 124) = 1.02, p = 0.400$ |

*Note.* * $p < 0.05$ two-sided; ** $p < 0.01$ two-sided; *** $p < 0.001$ two-sided.

Based on statistical significance and positive expected directions of relation, we accepted H2b, i.e., negative femininity is positively associated with wishful identification with stereotypical female character. Surprisingly, we found a statistically significant, positive relationship between negative femininity and wishful identification with a non-stereotypical female character. Our prediction was that the relationship would be negative. Hence, we rejected H3b. No other associations were statistically significant. Hence, H2a, H2c, H2d, H3a, H3c, H3d, H4a, H4b, H4c, H4d, H5a, H5b, H5c, and H5d were also rejected.

*4.2. Gendered Personality Traits and Parasocial Relationships*

4.2.1. Parasocial Friendship

A 2 (male character vs. female character) x 2 (stereotypical character vs. non-stereotypical character) ANOVA was conducted with parasocial friendship as a dependent variable. ANOVA revealed a significant main effect for the stereotypicality of the character, $F(1, 505) = 33.74, p < 0.001, \eta^2_p = 0.06$. Respondents experienced more parasocial friendship with non-stereotypical characters than with stereotypical characters. No significant effect was found for the gender of the character on the experience of parasocial friendship, $F(1, 504) = 1.75, p = 0.187, \eta^2_p < 0.01$. No significant interaction effect between the gender and the stereotypicality of the character in relation to the experience of parasocial friendship was found, $F(1, 504) = 2.11, p = 0.147, \eta^2_p < 0.01$. Therefore, we did not conduct comparisons in specific subgroups. Based on the negative direction of association found, H6a was declined because, against our expectations, parasocial friendships were lower for stereotypical characters. Moreover, no statistically significant results were found for the

effect of gender of the character. Hence, H6b was rejected. The summary of these results is presented graphically in Figure 2 and Table 3.

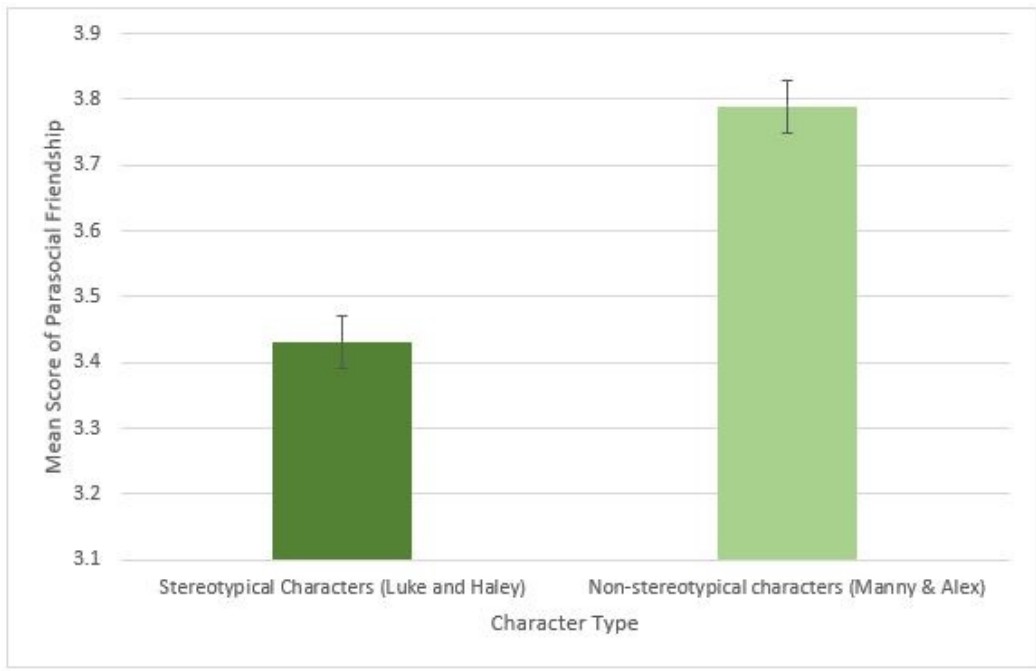

**Figure 2.** Mean scores of parasocial friendship with each type of character. *Note.* Error bars represent the standard error for mean scores.

**Table 3.** Means and standard deviations of parasocial friendship with various character groups.

| | Mean | Standard Deviation |
|---|---|---|
| Stereotypical characters (Haley and Luke) | 3.43 | 0.69 |
| Non-stereotypical characters (Alex and Manny) | 3.79 | 0.68 |
| Male characters (Luke and Manny) | 3.66 | 0.69 |
| Female characters (Alex and Haley) | 3.56 | 0.72 |
| Stereotypical female character (Haley) | 3.35 | 0.69 |
| Non-stereotypical female character (Alex) | 3.79 | 0.69 |
| Stereotypical male character (Luke) | 3.52 | 0.67 |
| Non-stereotypical male character (Manny) | 3.78 | 0.68 |

For each of the four *Modern Family* characters, one hierarchical regression analysis was conducted using parasocial friendship as a criterion. Predictors were the gendered personality traits added in the first block and wishful identification added in the second block. A summary of this analysis can be found in Table 4.

**Table 4.** Standardized beta coefficients for hierarchical regression analyses predicting parasocial friendship with studied characters.

| | Model 1 | Model 2 |
|---|---|---|
| Stereotypical female | | |
| (≠H7a) Positive femininity | 0.04 | −0.02 |
| (=H7b) Negative femininity | 0.26 ** | 0.18 * |
| (≠H7d) Positive masculinity | 0.16 | 0.18 * |
| (≠H7e) Negative masculinity | 0.07 | 0.04 |
| (=H7c) Wishful identification | | 0.40 *** |
| | $R^2 = 0.11$ | $\Delta R^2 = 0.14$ |
| | $F (4, 136) = 4.13$, $p = 0.003$ | $\Delta F (1, 135) = 25.21$, $p < 0.001$ |
| Non-stereotypical female | | |
| (≠H8a) Positive femininity | 0.17 | 0.16 |
| (≠H8b) Negative femininity | 0.17 | 0.06 |
| (≠H8c) Positive masculinity | −0.04 | −0.05 |
| (≠H8d) Negative masculinity | −0.15 | −0.09 |
| (=H8e) Wishful identification | | 0.50 *** |
| | $R^2 = 0.11$ | $\Delta R^2 = 0.23$ |
| | $F (4, 119) = 3.64$, $p = 0.008$ | $\Delta F (1, 118) = 42.08$, $p < 0.001$ |
| Stereotypical male | | |
| (≠H9a) Positive femininity | 0.07 | 0.15 |
| (≠H9b) Negative femininity | 0.01 | −0.08 |
| (≠H9c) Positive masculinity | 0.09 | 0.02 |
| (≠H9d) Negative masculinity | −0.02 | −0.03 |
| (=H9e) Wishful identification | | 0.55 *** |
| | $R^2 = 0.01$ | $\Delta R^2 = 0.29$ |
| | $F (4, 109) = 0.37$, $p = 0.829$ | $\Delta F(1, 108) = 45.58$, $p < 0.001$ |
| Non-stereotypical male | | |
| (≠H10a) Positive femininity | 0.14 | 0.11 |
| (≠H10b) Negative femininity | 0.10 | 0.04 |
| (≠H10d) Positive masculinity | 0.14 | 0.19 * |
| (≠H10e) Negative masculinity | 0.03 | 0.01 |
| (=H10c) Wishful identification | | 0.55 *** |
| | $R^2 = 0.07$ | $\Delta R^2 = 0.29$ |
| | F (4, 124) = 2.19, 9 = 0.074 | $\Delta F (1, 123) = 55.55$, $p < 0.001$ |

*Note.* * $p < 0.05$ two-sided; ** $p < 0.01$ two-sided; *** $p < 0.001$ two-sided.

Based on statistically significant results and the expected positive direction of association found, we accepted hypothesis H7b, i.e., parasocial friendship with a stereotypical female character is positively associated with negative femininity. Moreover, the association between wishful identification and parasocial friendship with a stereotypical female character, a non-stereotypical female character, a stereotypical male character, and a non-stereotypical male character were positive and statistically significant. Hence, we accepted H7c, H8e, H9e, and H10c. Surprisingly, we found a positive and statistically significant association between positive masculinity and parasocial friendship with a stereotypical female character and a non-stereotypical male character, which was against our assumption that the relationships would be negative. Therefore, we rejected H7d and H10d. All other associations were not statistically significant. Thus, H7a, H7e, H8a, H8b, H8c, H8d, H9a, H9b, H9c, H9d, H10a, H10b, and H10e were rejected.

### 4.2.2. Parasocial Love

A 2 (male character vs. female character) x 2 (stereotypical character vs. non-stereotypical character) ANOVA was conducted with parasocial love as a dependent variable. ANOVA revealed a significant main effect for the gender of the character, $F(1, 504) = 7.57$, $p$ 0.006, $\eta^2_p = 0.02$. Participants experienced more parasocial love with female characters than with male characters. A main effect for the stereotypicality of the character was not significant, $F(1, 504) = 0.01$, $p = 0.945$, $\eta^2_p < 0.001$. A significant interaction effect was found combining the effect of the gender of the character and the stereotypicality of the character on the experience of parasocial love, $F(1, 504) = 7.19$, $p = 0.008$, $\eta^2_p = 0.01$. Looking first at the (non-)stereotypical characters, post hoc $t$-tests revealed that participants experienced more parasocial love with the non-stereotypical female character than with the non-stereotypical male character, $t(230.10) = 3.82$, $p < 0.001$. Reviewing the experience of parasocial love with stereotypical characters, a post hoc $t$-test showed no differences between parasocial love with the stereotypical male character and with the stereotypical female character, $t(253) = 0.05$, $p = 0.961$. Further comparing experiences of parasocial love with female characters only, there were no statistically significant differences between the stereotypical female character and the non-stereotypical female character, $t(238.34) = 1.91$ $p = 0.057$. Similarly, when comparing feelings of parasocial love with male characters only, there were no statistically significant differences between the stereotypical male character and the non-stereotypical male character, $t(241) = 1.86$ $p = 0.064$. Based on the lack of statistical significance, H11a was rejected. Furthermore, H11b was rejected because of the negative direction of association found. Against our expectations, parasocial love was lower for male characters. The summary of these results is presented graphically in Figure 3 and Table 5.

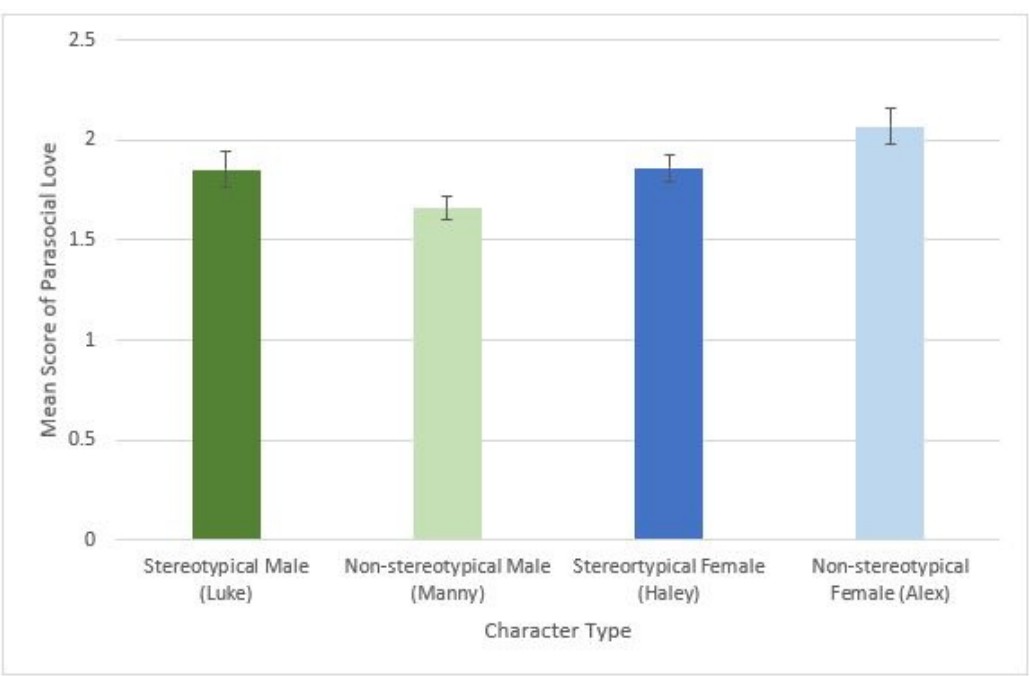

**Figure 3.** Mean scores of parasocial love with each type of character. *Note*. Error bars represent the standard error for mean scores.

**Table 5.** Means and standard deviations of parasocial love with various character groups.

|  | Mean | Standard Deviation |
|---|---|---|
| Stereotypical characters (Haley and Luke) | 1.86 | 0.85 |
| Non-stereotypical characters (Alex and Manny) | 1.86 | 0.88 |
| Male characters (Luke and Manny) | 1.75 | 0.83 |
| Female characters (Alex and Haley) | 1.96 | 0.88 |
| Stereotypical female character (Haley) | 1.86 | 0.79 |
| Non-stereotypical female character (Alex) | 2.07 | 0.96 |
| Stereotypical male character (Luke) | 1.85 | 0.92 |
| Non-stereotypical male character (Manny) | 1.66 | 0.74 |

For each of the four *Modern Family* characters, one hierarchical regression analysis was conducted using parasocial love as a criterion. Predictors were the gendered personality traits added in the first block and wishful identification added in the second block. A summary of this analysis can be found in Table 6.

**Table 6.** Standardized beta coefficients for *hierarchical* regression analyses predicting parasocial love with studied characters.

|  | Model 1 | Model 2 |
|---|---|---|
| **Stereotypical female** |  |  |
| ($\neq$H12a) Positive femininity | 0.01 | −0.06 |
| ($\neq$H12b) Negative femininity | 0.13 | 0.04 |
| ($\neq$H12d) Positive masculinity | 0.11 | 0.102 |
| ($\neq$H12e) Negative masculinity | 0.12 | 0.09 |
| (=H12c) Wishful identification |  | 0.42 *** |
|  | $R^2 = 0.05$ $F(4, 136) = 1.74, p = 0.143$ | $\Delta R^2 = 0.16$ $\Delta F(1, 135) = 27.25, p < 0.001$ |
| **Non-stereotypical female** |  |  |
| ($\neq$H13a) Positive femininity | 0.01 | 0.01 |
| ($\neq$H13b) Negative femininity | 0.19 | 0.09 |
| ($\neq$H13c) Positive masculinity | −0.12 | −0.13 |
| ($\neq$H13d) Negative masculinity | 0.03 | 0.10 |
| (=H13e) Wishful identification |  | 0.44 *** |
|  | $R^2 = 0.06$ $F(4, 119) = 1.87, p = 0.120$ | $\Delta R^2 = 0.21$ $\Delta F(1, 118) = 28.53, p < 0.001$ |
| **Stereotypical male** |  |  |
| ($\neq$H14a) Positive femininity | <0.001 | 0.08 |
| ($\neq$H14b) Negative femininity | 0.18 | 0.09 |
| ($\neq$H14c) Positive masculinity | 0.08 | 0.01 |
| ($\neq$H14d) Negative masculinity | 0.12 | 0.11 |
| (=H14e) Wishful identification |  | 0.54 *** |

**Table 6.** *Cont.*

|  | Model 1 | Model 2 |
|---|---|---|
|  | $R^2 = 0.06$ | $\Delta R^2 = 0.28$ |
|  | $F (4, 109) = 1.71, p = 0.153$ | $\Delta F(1, 108) = 45.43, p < 0.001$ |
| Non-stereotypical male |  |  |
| ($\neq$H15a) Positive femininity | 0.01 | $-0.02$ |
| (=H15b) Negative femininity | 0.23 * | 0.17 * |
| ($\neq$H15d) Positive masculinity | 0.03 | 0.08 |
| ($\neq$H15e) Negative masculinity | 0.13 | 0.11 |
| (=H15c) Wishful identification |  | 0.55 *** |
|  | $R^2 = 0.07$ | $\Delta R^2 = 0.29$ |
|  | $F (4, 124) = 2.26, p = 0.066$ | $\Delta F (1, 123) = 56.10, p < 0.001$ |

*Note.* * $p < 0.05$ two-sided; ** $p < 0.01$ two-sided; *** $p < 0.001$ two-sided.

Based on statistically significant results and the expected positive direction of association found, H15b can be accepted: negative femininity is positively associated with parasocial love with a non-stereotypical male character. Moreover, we found statistically significant and positive associations between wishful identification and parasocial love with stereotypical female, non-stereotypical female, stereotypical male, and non-stereotypical male characters. Hence, H12c, H13e, H14e, and H15c were accepted. No other associations were statistically significant. Hence, H12a, H12b, H12d, H12e, H13a, H13b, H13c, H13d, H14a, H14b, H14c, H41d, H15a, H15d, and H15e were rejected

### 4.3. Enjoyment of Modern Family

A multiple regression was conducted using enjoyment as a criterion and the gendered personality traits as predictors. A summary of the results can be found in Table 7.

**Table 7.** Standardized beta coefficients for hierarchical regression analyses predicting enjoyment.

| | |
|---|---|
| (=H16a) Positive femininity | 0.21 *** |
| ($\neq$H16b) Negative femininity | $-0.06$ |
| ($\neq$H16c) Positive masculinity | 0.02 |
| ($\neq$H16d) Negative masculinity | $< -0.01$ |
|  | $R^2 = 0.04$ |
|  | $F (4, 503) = 4.54, p = 0.001$ |

*Note.* * $p < 0.05$ two-sided; ** $p < 0.01$ two-sided; *** $p < 0.001$ two-sided.

Based on the statistical significance and the positive expected direction of association, we accepted H16a, i.e., enjoyment is positively associated with positive femininity. No other associations were statistically significant. Therefore, H16b, H16c, and H16d were rejected.

## 5. Discussion

In this research, we investigated to what extent gendered personality traits relate to various media engagement measurements. Our two goals were (1) to verify that gendered personality traits are valuable predictors when studying relationships that are built with TV characters and (2) to analyze how the characters' (non-)stereotypical gender portrayal influences the relationships between audience members and media figures.

Responding to our first research goal, we have found that gendered personality traits can indeed predict various ways of engagement with media characters and TV shows. First, negative femininity was positively associated with wishful identification and with parasocial friendship with a stereotypical female character, Haley. This result could be explained by some personality overlap between negative femininity and stereotypical female media portrayal. Audience members who scored high on negative femininity

reported being oversensitive, overcautious, and anxious. Haley is a popular and friendly character. Interestingly, both sets of traits share a community orientation, either through rejecting the focus on the self and sensitivity or through seeking contact with others. Moreover, people with strong negative feminine traits could wish to develop the more positive, empathetic community orientation shown by Haley. Second, negative femininity was associated with parasocial love with a non-stereotypical male character Manny, who is portrayed as sensitive, passionate, and also anxious at times. These very same traits also characterized people scoring high on negative femininity. Thus, this result could be explained by personality similarity, which could make the character more familiar, understandable, and ultimately lovable. Additionally, our sample was predominantly female, which could explain the romantic preference for the male character. Third, we also found that positive femininity predicted enjoyment of *Modern Family*. This result is not surprising as the series showcases different families. Multiple characters show community orientation, which is a crux of positive femininity, which embraces being loving, emphatic, and emotional. Therefore, it is possible that the viewers who scored high on positive femininity created multiple parasocial bonds with different characters from various families that were not studied in this project, which could explain their greater enjoyment of the show.

In general, such results showing the importance of similarities between viewers and characters for wishful identification and parasocial relationships have been reported before (Cohen and Hershman-Shitrit 2017; Hoffner and Buchanan 2005; Tukachinsky et al. 2020). Furthermore, enjoyment of TV series stemming from parasocial relationships with similar characters has also been previously reported (Baldwin and Raney 2021; Kim and Sintas 2021; Rosaen and Dibble 2017). As a novel contribution, we found empirical support for the previously understudied possible dimension of similarity, namely gendered personality traits. The discussed relationships we found offer support to our claim that viewers can find similarities with characters beyond simple demographic identity dimensions but rather relying on more complex identity constructs. Moreover, these similarities can promote engagement with media characters. For instance, it is possible for predominantly Western viewers with high negative femininity to feel parasocial love for a Latino character, Manny. Presumably, it is the similarity of personality that won over the possible differences in racial backgrounds and experiences, which ultimately explains this connection. Mora (2018) reported similar findings and argued that Gloria Pritchett, a Latina character, is well liked by white audience members.

Some of our results were against our expectations but still show links between gender personality traits and media engagement. First, negative femininity was positively associated with wishful identification with a non-stereotypical female character. This relationship cannot be explained by similarity of attitudes, as reported by Hoffner and Buchanan (2005). Viewers who self-reported to be oversensitive and self-doubting also wanted to be more similar to Alex from *Modern Family*, an independent, smart, and non-stereotypical female character. Perhaps viewers wanted to be similar to Alex because they wanted to develop more independence and self-reliance in their own lives. Wishful identification can possibly depend on individual needs of the viewer. However, this claim needs further exploration and testing in rigorous qualitative and quantitative studies. Second, we found that viewers who scored higher on positive masculinity also reported higher levels of parasocial friendship with a stereotypical female character, Haley. This result is also hard to explain when thinking of gendered traits' similarity. Analytical and solution-oriented respondents (i.e., with high positive masculinity) felt close to the character who is passionate, emotional, and more community driven. Perhaps viewers wanted to develop more community orientation because they lacked it in their own lives. Moreover, Haley's popularity could be because she is generally attractive. Attractive character representation could also be a predictor of parasocial interactions (Tukachinsky et al. 2020). Finally, we also did not find statistically significant results for the 45 hypotheses we posed. This means that viewers scoring high on personality traits shared with a given character did not develop stronger bonds with this

character when compared to viewers who scored low on the overlapping personality traits. Still, we found generally high scores on wishful identification and parasocial friendships for all types of characters (see Figures 1 and 2), which means that media engagement is possible despite differences in personality.

In general, these unexpected findings complicate our understanding of similarity and media engagement. It is likely that personality similarity is not a necessary factor for promoting engagement but rather a contributing one. Therefore, television shows should not shy away from showcasing a variety of individuals that initially do not seem familiar to the target audiences. Still, the role of perceived similarity cannot be completely dismissed as the concept predicted media engagement in other research (Cohen and Hershman-Shitrit 2017; Hoffner and Buchanan 2005; Tukachinsky et al. 2020) and explained some of the media engagement patterns we have found. Moreover, it is also possible that viewers engage with characters because of sharing similarities that were not measured in our research. Perhaps the similarity of particular life experiences or situations that could be connected to other identity aspects, such as age, social class, race, ethnicity, class, or religion, can predict different media engagement patterns that we have not found due to our research focus. Lastly, it is possible that engagement with characters depends on more factors, such as attractiveness and likability (Tukachinsky et al. 2020) or other factors, which should be studied in future explorative research projects.

As another result, we found that different ways of engaging with media could go hand in hand. Wishful identification predicted parasocial friendship and love for all the studied characters, no matter their gender or stereotypical portrayal. These conclusions are in line with previous literature (Hu et al. 2021; Lim et al. 2020).

Responding to our second research goal, we also examined how the gender of the characters and stereotypical portrayal affected media engagement. We found that wishful identification is higher for non-stereotypical and female characters, parasocial friendships were higher for non-stereotypical characters, and parasocial love was higher for female characters. These findings go *against* the seminal piece by Perse and Rubin (1989), who argued that uncertainty reduction is important in bonding with media characters. It seems that the needs of the contemporary audience have changed when engaging with recent shows. Instead of uncertainty reduction via stereotypical characters, non-stereotypical characters seem to be more interesting, as the viewers want to be similar to them and develop more friendly feelings for them. As mentioned, the attractiveness of characters is also a known predictor of media engagement. However, there is still little agreement on how to achieve attractive representation (Żerebecki et al. 2021). Perhaps media characters defying stereotypes is one of the ways to achieve attractive representation. Moreover, the non-stereotypical characters in question, Alex and Manny, are portrayed as more successful concerning education and business, which possibly also makes them more attractive to viewers. Considering that the majority of our sample were young females, we can argue that such characters serve as new and modern role models and, thus, connection to them is important.

Based on all our results, it can be concluded that a mix of stereotypical and non-stereotypical characters in TV shows is important to safeguard a diverse audience base. While we found that, in general, non-stereotypical characters elicited more media engagement, we also found specific higher levels of parasocial love and wishful identification with stereotypical characters for specific subgroups of our sample.

*5.1. Limitations and Suggestions for Future Research*

This study has several limitations. First, data were collected cross-sectionally. Therefore, we cannot infer any causal relationships with certainty. While it is likely that gendered personality traits are stable over time, future research should examine this in a longitudinal design. Perhaps TV shows can cultivate certain personality traits in susceptible viewers, making them exhibit more gendered personality traits of a given type over time.

Second, the survey was distributed to respondents via non-random sampling. Although the survey information was posted on relevant fan groups on social media, which resulted in an international composition of the sample, we still had predominantly Western participants. In particular, the Netherlands, the United Kingdom of Great Britain and Northern Ireland, and the United States of America were most common. Considering the international success and audience of *Modern Family* (Zeitchik 2019), a more international audience would have been preferable. This is a significant limitation since cultural background can affect an individual's parasocial relationships and wishful identification with a TV character (Mora 2018). A similar cultural background can positively impact the connections made with a character. Most of the participants have a similar background as the four characters, of which three are American and one is Latin American. Future research projects could rectify this by using quota sampling to ensure a balance between Western and non-Western participants.

Third, *Modern Family* is an American production, which underlines the Western focus of this study. Analyzing a non-Western TV show might give different results. Moreover, the used gendered personality traits also come from a Western background (Kneer et al. 2019). Non-Western cultures possibly have different views on what personality traits are feminine or masculine. The view on gender stereotypes and gender non-stereotypes is also perceived differently among different cultures (Wood and Eagly 2009). Thus, our findings are applicable mostly within the West. Future research could use different measures of gendered personality traits.

Fourth, our stimulus material included three white characters (Alex, Haley, and Luke) and one Latino character (Manny). This choice was a necessary one due to a lack of an equivalent, non-stereotypical, white, male character but could have affected our results. Our respondents reported low levels of parasocial love for Manny, thus possibly showing evidence of prejudice toward Latino men in this respect. We designed this experiment with real-life materials, which limited our chances to select stimulus material with manipulated portrayal of the characters. Future studies could produce fictional materials where characters share the same racial background to check whether our results can be replicated in such a setting.

Fifth and last, the unbalanced gender composition of this project has to be considered when drawing conclusions. While we included only gendered personality traits in the analyses, it is possible that our findings could differ if we had more male-identifying participants. Still, our sample included respondents who displayed various personality traits. Moreover, the omission of binary gender as a variable enabled us to take into account the responses of non-binary people and people who did not disclose their gender. Future research could strive for the inclusion of people with various gender identities and try to balance out the composition of the sample through quota sampling.

*5.2. Practical Conclusions*

Despite our limitations, our research contributes to both society and the academic world. We have shown that gender does not have to be conceptualized as a single-item variable but can be thought of in an intersectional manner, which embraces identity as a multifaceted concept. Moreover, based on our findings, several practical implications can be proposed. For media producers and scriptwriters, we recommend a diverse set of characters, consisting of male, female, masculine, feminine, gender-stereotypical, and gender-non-stereotypical people. Different gendered personality traits predict relationships with different types of characters. Therefore, to reach a broad and diverse audience, it is important to have a cast that provides potential connections for all types of viewers. Media characters should exhibit both stereotypical and non-stereotypical traits to make sure they are likable to broad sections of the audience members. Moreover, viewers can create bonds with media characters that are not similar at first glance (due to different gender expression or race) or even with characters that do not have similar personalities. This means that diverse characters should not be seen as catering only to minority audiences but

rather as characters who can be engaging and liked by all kinds of viewers. Our research shows that similarity should not be thought of in simple demographic terms but rather as a multi-faceted construct that can connect seemingly different people. For scholars, we recommend a continued focus on exploring gender personality traits as predictors of media use and engagement. Furthermore, we would like to urge other scholars to think about different ways in which a sense of similarity despite visible differences in class, race, ethnicity, religion, or sexual orientation can be captured. We hope that more research in this area will help us understand the ever-changing experiences provided by media engagement with different characters of audience members coming from social minority and majority groups.

**Author Contributions:** Conceptualization, E.v.d.V. and J.K.; methodology, E.v.d.V. and J.K.; formal analysis, B.G.Ż., E.v.d.V. and J.K.; writing—original draft preparation, B.G.Ż., E.v.d.V. and J.K.; writing—review and editing, B.G.Ż., E.v.d.V. and J.K.; visualization, B.G.Ż. and J.K.; supervision, J.K. All authors have read and agreed to the published version of the manuscript.

**Funding:** This research received no external funding.

**Informed Consent Statement:** Informed consent was obtained from all subjects involved in the study.

**Data Availability Statement:** The data presented in this study are available on request from the corresponding author. The data are not publicly available due to privacy reasons and ethical concerns.

**Conflicts of Interest:** The authors declare no conflict of interest.

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
