# Peer review of "I Want to Be You(r Friend): An Investigation of the Effects of Gendered Personality Traits on Engagement with Different Modern Family Characters"

_journalmedia, doi:10.3390/journalmedia3020026_

Round 1

Reviewer 1 Report

The research takes us a step closer in understand what about gender-similarity can produce nuanced parasocial interactions with TV characters. But, the one lingering question remains: why was gender chosen over other identity categories like ethnicity/race, class, or religion? They argue in the second paragraph in the introduction that audiences have become more diversity, and then proceed to talk about gender as an example, and they keep on talking about gender in subsequent paragraphs without providing a case as to why gender is the best identity category to focus their research. The author(s) do a great job describing the diversity of gender later in the literature review, but I would stress that they should explain why gender makes a better case than other categories in the introduction.

The author(s) frame their intent to take us deeper into gender through a lens of intersectionality (p. 2). They acknowledge that intersectionality is about overlapping identities, which I agree, but they only talk about gender in positive or negative terms without really explaining how these overlap with other identities. I would urge the author(s) to either explain why the negative/positive femininity/masculinity counts as intersectionality, or how gender and ethnicity for example, come together to create a certain trait among characters to really stay true to intersectionality, or should find another theoretical framework that better accounts their zooming into gender.    

The literature review and hypothesis are solid and well-researched and summarized- kudos to the author(s).

In the methods section, I would just like to read a sentence as to how each character chosen for the stimulus has a positive/negative femininity/masculinity in order to create a picture for the reader in the case they have a vague understanding of the show. It is also worth noting that most of these stimuli characters were the youngsters in the show. There needs to be some justification as to why older characters were not selected. This has serious implications through the paradigm of intersectionality as age and gender and in the case of Manny Delgado additionally has ethnicity that further complicates how TV audiences may react to him.

As I read the discussion section, I think a better explanation of the characters is needed to add context and better connections, especially with the amount of hypotheses the study has. I found myself re-reading the paragraphs several times trying to see how and why results went a certain way. I understand that the overall goal of the study was to see/show that gender traits correlates to media engagement. Trying to include more details about the characters and the gender traits can lead to direct connections and for the audience to draw specific mental pictures. Most of the earlier explanations in the discussion were attributed to overlaps. I do see here an opportunity for narrative contextualization or other writing techniques that give some more details. On page 18, towards the bottom, the authors do a great job writing how Alex and Manny could be role models and this gave a clear explanation. They can use this as the example to write more about the characters and statistical results observed.

On page 18, the author(s) write: “As a novel contribution, we found empirical support for the previously understudied possible dimension of similarity, namely gendered personality traits. The discussed relationships we found offer support to our claim that viewers can find similarities with characters beyond simple demographic identity dimensions but rather relying on more complex identity constructs. Moreover, these similarities can promote engagement with media characters.” I don’t argue against the contribution and observations of the study. This is a great opportunity to be critical of the work about what this means in the larger discussion of gender in the media. The author(s) cite Mora (2018) who found that non-Latinos found the character of Gloria Pritchett just as charming as Latino TV viewers despite being a stereotypical Latina character. Again, the author(s) could engage in a critical discussion with the literature that exists out there about Modern Family TV audiences for us to see/read the contribution to this reception of the show and what it means to interact with characters who may or not share gender AND ethnic similarity.    

Reviewer 2 Report

I think this is really interesting research, especially given that so many of your hypotheses were not supported. I always believe that the most interesting findings come when we are wrong in our guesses. I would have liked to have seen you dive deeper into why your hypothesis might not have been supported. This is obviously challenging when doing a purely quantitative study, but I think a critique of our current societal standings in relation to your findings is valid. Overall, I am excited about this research and the exploration of how viewers identify with characters that don't match our traditional expectations of gender.
